# Prompt Backdoors in Visual Prompt Learning

## Abstract

Fine-tuning large pre-trained computer vision models is infeasible for resource-limited users. Visual prompt learning (VPL) has thus emerged to provide an efficient and flexible alternative to model fine-tuning through Visual Prompt as a Service (VPPTaaS). Specifically, the VPPTaaS provider optimizes a visual prompt given downstream data, and downstream users can use this prompt together with the large pre-trained model for prediction. However, this new learning paradigm may also pose security risks when the VPPTaaS provider instead provides a malicious visual prompt. In this paper, we take the first step to explore such risks through the lens of backdoor attacks. Specifically, we propose BadVisualPrompt, a simple yet effective backdoor attack against VPL. For example, poisoning $5\%$ CIFAR10 training data leads to above $99\%$ attack success rates with only negligible model accuracy drop by $1.5\%$. In particular, we identify and then address a new technical challenge related to interactions between the backdoor trigger and visual prompt, which does not exist in conventional, model-level backdoors. Moreover, we provide in-depth analyses of seven backdoor defenses from model, prompt, and input levels. Overall, all these defenses are either ineffective or impractical to mitigate our BadVisualPrompt, implying the critical vulnerability of VPL.[1]

## 1 Introduction

Large pre-trained computer vision models have shown great success in various (downstream) tasks (Aygun et al., 2017; Dosovitskiy et al., 2021; Fang et al., 2021; Zhuang et al., 2021). However, the conventional approach to adapting pre-trained models based on fine-tuning model parameters requires large computations and memories. To address such limitations, inspired by the success of prompt learning in NLP (Shin et al., 2020; Hambardzumyan et al., 2021; Lester et al., 2021; Li & Liang, 2021), recent work has introduced visual prompt learning (VPL) (Bahng et al., 2022; Gan et al., 2023; Wu et al., 2022; Chen et al., 2023a;b; Huang et al., 2023) as an efficient alternative for model adaption. Given a pre-trained visual model and specific downstream data, VPL learns a global visual prompt, which comprises pixel perturbations, usually in the shape of a padding or patch. This learned visual prompt is then placed on any downstream test image for model prediction. Recent work has shown the competitive performance of VPL to parameter fine-tuning in various tasks (Bahng et al., 2022; Gao et al., 2022; Liu et al., 2022).

A suitable visual prompt is critical to the VPL performance and normally needs considerable efforts to optimize (Bahng et al., 2022; Gan et al., 2023; Wu et al., 2022; Chen et al., 2023a;b). Therefore, Visual Prompt as a Service (VPPTaaS) is promising to assist non-expert users to adapt to this new paradigm, as in the NLP domain (Phr; Ding et al., 2022). In a typical scenario, users provide their data to the VPPTaaS provider to optimize a prompt. Then, the prompt is returned to users and can be used together with a pre-trained visual model for prediction. However, despite its effectiveness and convenience, VPPTaaS may bring unknown security risks to downstream users when the VPPTaaS provider intentionally supplies a malicious visual prompt.

In this paper, we take the first step to systematically study the security risks of VPPTaaS. We focus on the backdoor attack since it is widely recognized as a major security risk of machine learning models (Gu et al., 2017; Jia et al., 2022a). Specifically, we propose BadVisualPrompt, the first backdoor attack against VPL. Different from conventional backdoors, which are implanted in model parameters, our backdoor is implanted in the (pixel-space) visual prompt. With such a backdoored

---

[1]Our code is available at https://anonymous.4open.science/r/BadVisualPrompt.

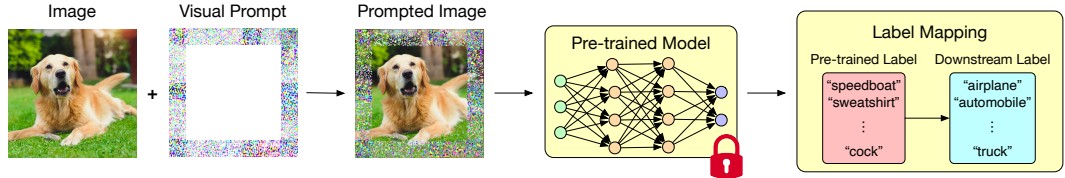

Figure 1: Illustration of applying a visual prompt in visual prompt learning (VPL). An input image is combined with the learned visual prompt and then sent to the fixed pre-trained model for prediction. Label mapping is used for addressing the difference between the upstream and downstream tasks. Note that the original image should be first resized to the desired input size of the pre-trained model.

visual prompt, the pre-trained model would behave abnormally (e.g., misclassifying the input) when a pre-defined backdoor trigger appears in the input but normally on a clean input.

Our systematic studies are conducted from both the attack and defense perspectives. From the attack perspective, we demonstrate the effectiveness of BadVisualPrompt in various settings with diverse model architectures, datasets, and VPL variants (with different prompt templates or label mapping strategies). For instance, poisoning $5\%$ CIFAR10 training data leads to an average attack success rate (ASR) of above $99\%$ with only about a $1.5\%$ drop in model clean accuracy (CA). In particular, we point out that trigger-prompt interactions should be studied since both the trigger and prompt are placed on the same input image. As a case study, we analyze the impact of trigger-prompt distance on the attack performance and find that the ASR may drop by $80\%$ when the trigger appears distant from the visual prompt. We further show that optimizing the trigger pattern can restore the ASR in this challenging case.

From the defense perspective, we provide in-depth analyses of seven backdoor detection and mitigation methods from three different levels: model, prompt, and input. In general, we find that these defenses are either ineffective or impractical against our new attack, BadVisualPrompt. In particular, we investigate a new, prompt-level detection method that is based on visual discrimination of backdoored and clean prompts. We find that although this new prompt-level detection method achieves almost $100\%$ accuracy, a large number of training prompts and substantial computational resources are required.

Note that, the major contribution of this paper is not proposing new attack techniques, but systematically and empirically evaluating the security risks of VPL, a brand new learning paradigm for large vision models. Our work provides significant findings and in-depth analysis which might inspire further security research in VPL.

## 2 BADVISUALPROMPT

### 2.1 VISUAL PROMPT LEARNING

Recall that pixel space (Bahng et al., 2022) is continuous. Inspired by the continuous prompts (sometimes called soft prompts in NLP), visual prompt learning (Bahng et al., 2022) aims at learning a visual prompt $\mathbf{w}$ to adapt the input image $\mathbf{x}$ to a pre-trained image model $M$ (see Figure 1 for illustration). The downstream task learns a function $f(M, \mathbf{w}, \mathbf{x})$ that combines the frozen pre-trained model $M$ and the visual prompt $\mathbf{w}$ to predict the result for an input image $\mathbf{x}$. Concretely, the visual prompt is optimized on the downstream training dataset $\mathcal{D}$ with the following objective function:

$$\mathbf{w}^* = \arg\min_{\mathbf{w}} \mathbb{E}_{(\mathbf{x},y)\in\mathcal{D}}[\mathcal{L}(f(M, \mathbf{w}, \mathbf{x}), y)], \tag{1}$$

where the loss function $\mathcal{L}(\cdot)$ is normally the cross-entropy loss. If the downstream task is a classification task, visual prompt learning also requires a pre-defined label mapping $\pi$ to interpret the prompting results (see Chen et al. (2023b)). Moreover, a visual prompt can be seen as additive perturbations to the input image. Users may use any form of visual templates (e.g., patch and padding) to represent visual prompts in practice. Note that one recent study (Jia et al., 2022b) adopts a different paradigm that forms the prompt as additional model parameters instead of pixel perturbations at the input space. This paradigm and its variants are therefore out of our research scope. A detailed review of related work on (visual) prompt learning and backdoor attacks/defenses can be found in Appendix A.

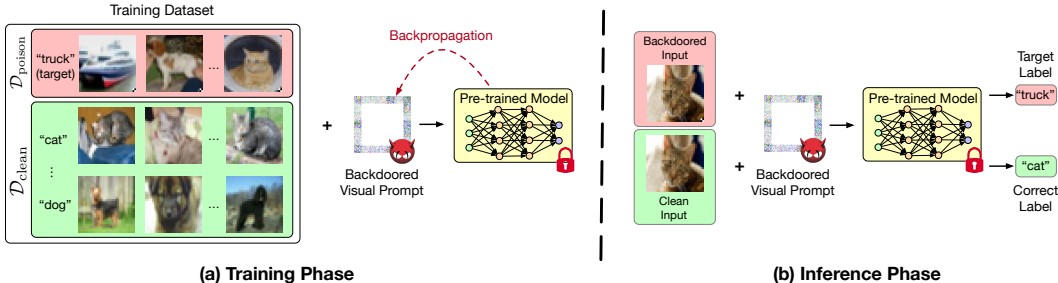

Figure 2: Workflow of BadVisualPrompt. In the (a) training phase, the visual prompt is optimized on clean and poisoned data to contain backdoor information. Then, in the (b) inference phase, the backdoored prompt can be applied to triggered images for targeted misclassification.

## 2.2 THREAT MODEL

Following existing backdoor studies (Gu et al., 2017), we assume that the attacker is a malicious VPPTaaS service provider. The victim, i.e., the downstream user, outsources the prompt optimization to the VPPTaaS provider and may get back a backdoored visual prompt. We assume the pre-trained model is publicly available (to both the attacker and victim). **Attacker's Goals.** The attacker aims to implant backdoors in the visual prompt. When such a backdoored visual prompt is returned to the victim, their downstream prediction is correct for clean inputs but incorrect for the triggered inputs. As such, the attacker tends to simultaneously achieve two goals, i.e., achieving attack success and maintaining model utility. **Attacker's Knowledge and Capabilities.** To get a task-specific visual prompt, the user must supply detailed downstream task information, including limited downstream data, to the service provider. Therefore, we assume that the attacker has knowledge of the downstream dataset. We also assume that the attacker has full control of the prompt learning process and can define the form of the visual prompt (e.g., shape and location).

## 2.3 ATTACK METHOD

**Data Poisoning.** Our attack method, namely BadVisualPrompt, crafts a backdoored visual prompt by manipulating the user-uploaded dataset (denoted as $\mathcal{D}_{\text{clean}}$) in the prompting process. We randomly sample a proportion of $p$ from $\mathcal{D}_{\text{clean}}$ to constitute a poisoned dataset $\mathcal{D}_{\text{poison}}$. Specifically, for each sampled instance $(\mathbf{x}, y) \in \mathcal{D}_{\text{clean}}$, we form its corresponding poisoned version $(\mathbf{x}_{\text{poison}}, t) \in \mathcal{D}_{\text{poison}}$, where $\mathbf{x}_{\text{poison}} = \mathcal{P}(\mathbf{x}, \Delta, t)$ and $\mathcal{P}(\cdot)$ is a function to add the backdoor trigger $\Delta$ to the given image $\mathbf{x}$ and to assign an incorrect, target label $t$. For the trigger $\Delta$, following the common practice (Gu et al., 2017), we adopt a small patch with iterative white and black colors placed at the corner.

**Attack Objective.** The optimization of BadVisualPrompt can be formulated as:

$$\mathbf{w_b} = \arg\min_{\mathbf{w}} \big[ \mathbb{E}_{(\mathbf{x},y) \in \mathcal{D}_{\text{clean}}} \mathcal{L}(f(M, \mathbf{w}, \mathbf{x}), y) + \lambda \cdot \mathbb{E}_{(\mathbf{x}_{\text{poison}},t) \in \mathcal{D}_{\text{poison}}} \mathcal{L}(f(M, \mathbf{w}, \mathbf{x}_{\text{poison}}), t) \big], \quad (2)$$

where the $\mathcal{L}(\cdot)$ represents the loss function (e.g., cross-entropy loss) in the normal prompting process, and $\lambda > 0$ is a coefficient to balance the model utility (i.e., first term) and attack effectiveness (i.e., second term). Intuitively, a larger $\lambda$ makes the backdoored visual prompt $\mathbf{w_b}$ focus more on the attack effectiveness and may exert a larger negative impact on the model utility.

**Workflow.** The workflow of our BadVisualPrompt is illustrated in Figure 2. In the training phase, we optimize the backdoored visual prompt using both $\mathcal{D}_{\text{poison}}$ and $\mathcal{D}_{\text{clean}}$. In the inference phase, the backdoored visual prompt $\mathbf{w_b}$ is placed on (clean or backdoored) images to feed into the pre-trained model. Specifically, the model can correctly classify the clean input but misclassify the triggered input, into a target class.

## 3 Experiments of Attacks

### 3.1 Experimental Setups

**Datasets and Models.** We consider three benchmark image datasets: CIFAR10 (CIF), SVHN (Netzer et al., 2011), and EuroSAT (Helber et al., 2018). We use the official training and testing data splits for the CIFAR10 and SVHN datasets. For the EuroSAT dataset, we randomly sample $80\%$ images per class for training and the rest $20\%$ for testing. For pre-trained models, we consider three vision models, i.e, ResNet trained on ImageNet-1K (RN50) (He et al., 2016; Russakovsky et al., 2015), Big Transfer (BiT-M) (Kolesnikov et al., 2020), ResNeXt trained on 3.5B Instagram images (Instagram) (Mahajan et al., 2018), and also a vision-language model, CLIP (Radford et al., 2021).

**Prompt Learning and Attack Settings.** We follow Bahng et al. (2022) to construct visual prompts. If not mentioned specifically, the visual prompt has a shape of four-edge padding with a width of 30 pixels (on a $224 \times 224$ input image). For label mapping, each pre-trained class index $i$ corresponds to the same downstream class index $i$ for the three vision models, and a semantically similar text prompt is constructed for CLIP. For attacking, we place the trigger at the bottom right corner with the size as 1/16 of the input image and set $\lambda = 1.0$ in Equation 2. We consider both single-target and multi-target attack goals. For the single-target goal, we choose "automobile" for CIFAR10, "1" for SVHN, and "forest" for EuroSAT, all mapping to class index 1, and we poison $5\%$ training data. For the multi-target goal, we choose class indexes 1, 3, and 5 for each dataset. We adopt different trigger positions for different targets (i.e., bottom left $\rightarrow$ 1, bottom center $\rightarrow$ 3, and bottom right $\rightarrow$ 5), and we poison $2\%$ training data for each target.

**Evaluation Metrics.** In this work, the pre-trained model is always used together with the visual prompt to give predictions for downstream data. We use Clean Accuracy (CA) and Attack Success Rate (ASR) to measure the performance of our BadVisualPrompt. Here CA represents the percentage of clean test images whose predicted labels are the same as the ground-truth labels, while ASR represents the percentage of backdoored test images whose predicted labels are the same as the target labels. In general, a higher ASR with little impact on CA indicates a more effective backdoor attack. More detailed descriptions of our experimental setups can be found in Appendix B.

### 3.2 Effectiveness of BadVisualPrompt

As can be seen from Table 1, our Bad-VisualPrompt achieves ASRs higher than 99% with less than 1% CA drop in most cases. The relatively low ASR (78.47%) of the RN50 model on EuroSAT for the multi-target attack is mainly caused by the low ASR (47.11%) for label index 5. We find that class index 5 has the minimum training samples (i.e., 1,600), so a model that generalizes not very well, i.e., the RN50 with a CA of 79.63%, may yield relatively low attack results.

Table 1: Single- and multi-target attack performance.

| Model | Prompt | Metric | Single-target attack | | | Multi-target attack | | |
|---|---|---|---|---|---|---|---|---|
| | | | CIFAR10 | EuroSAT | SVHN | CIFAR10 | EuroSAT | SVHN |
| RN50 | Clean | CA (%) | 54.99 | 79.63 | 60.95 | 54.99 | 79.63 | 60.95 |
| | | ASR (%) | 9.33 | 11.67 | 21.46 | 9.80 | 8.48 | 13.45 |
| | Backdoor | CA (%) | 54.75 | 79.33 | 59.91 | 54.29 | 80.19 | 59.41 |
| | | ASR (%) | 99.94 | 99.59 | 100.00 | 96.19 | 78.47 | 99.22 |
| BiT-M | Clean | CA (%) | 61.91 | 85.72 | 69.43 | 61.91 | 85.72 | 69.43 |
| | | ASR (%) | 10.56 | 10.94 | 20.18 | 9.54 | 9.64 | 14.69 |
| | Backdoor | CA (%) | 62.45 | 86.22 | 70.99 | 61.67 | 86.28 | 68.73 |
| | | ASR (%) | 100.00 | 100.00 | 100.00 | 99.96 | 99.40 | 99.86 |
| Ins. | Clean | CA (%) | 64.22 | 84.96 | 72.02 | 64.22 | 84.96 | 72.02 |
| | | ASR (%) | 9.91 | 11.20 | 20.84 | 10.79 | 9.35 | 13.39 |
| | Backdoor | CA (%) | 63.07 | 85.96 | 68.80 | 64.54 | 85.35 | 69.99 |
| | | ASR (%) | 99.50 | 99.94 | 99.90 | 96.84 | 95.33 | 98.77 |
| CLIP | Clean | CA (%) | 92.94 | 96.91 | 90.88 | 92.94 | 96.91 | 90.88 |
| | | ASR (%) | 9.93 | 10.98 | 20.10 | 9.95 | 9.25 | 13.52 |
| | Backdoor | CA (%) | 92.95 | 96.46 | 90.34 | 92.32 | 96.15 | 90.76 |
| | | ASR (%) | 99.99 | 99.94 | 100.00 | 99.80 | 98.95 | 99.95 |

We further test the impact of the poisoning ratio and trigger size on the attack performance. Detailed results for three datasets and four models are shown in Figure 11 of Appendix C. As expected, in general, the attack performance increases as the poisoning ratio or trigger size increases. Specifically, we find that the attack performance gets saturated for most cases even when the ratio is lower than $3\%$ or the trigger size is $2 \times 2$. Another interesting observation is that CLIP yields better attack performance than the vision models. One exception is when the trigger size is small. We attribute this finding to the high capacities of CLIP and provide detailed explanations about the particularly low results for EuroSAT in Appendix C.

### 3.3 New Insights into Trigger-Prompt Interactions

In conventional backdoor attacks, the impact of trigger position on the attack performance is negligible (Zeng et al., 2022; Liu et al., 2018c; Xu et al., 2021). However, in our context, changing the trigger position may lead to different interactions between the trigger and the visual prompt since they are placed on the same input image. As can be seen from Figure 3, our exploratory experiments show that placing the trigger in the central position leads to a significantly low ASR, except for CLIP.

To further analyze the impact of trigger-prompt interactions on the attack performance, we formulate the problem as gradually moving the trigger further away from the prompt, as illustrated in Figure 4. Here we choose a larger trigger size (i.e., $4 \times 4$) to capture more variances of the trigger-prompt overlap. The $4 \times 4$ trigger on the original $32 \times 32$ image is resized to $28 \times 28$ on the resized $224 \times 224$ image. In addition to the padding prompt, we consider a stripe prompt with the size of $30 \times 224$ and a patch prompt with the size of $80 \times 80$. We define the position of a trigger as $(h, w)$, where $h/w$ represents the vertical/horizontal coordinate pixel distance from the top left corner of the trigger to that of the resized image.

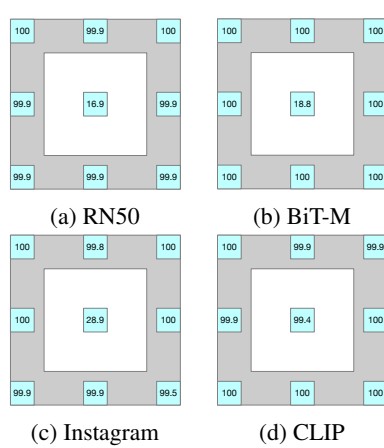

(a) RN50   (b) BiT-M

(c) Instagram   (d) CLIP

Figure 3: Attack success rates (%) of backdoored visual prompts (in gray color) with triggers (in blue color) at 9 typical positions on CIFAR10.

We measure the trigger-prompt interactions by their overlap and distance, which is their minimum pixel distance. As can be seen from Table 2, the trigger-prompt overlap has little impact on the attack performance. For example, both the CA and ASR results for padding remain almost the same when the overlap decreases from 784 to 108. In contrast, the trigger-prompt distance has a significant impact. For example, the ASR drops from $84.08\%$ to $17.76\%$ when the distance increases from 26 to 54. We find that the above observation also holds for the frequency-based label mapping strategy (Chen et al., 2023b).

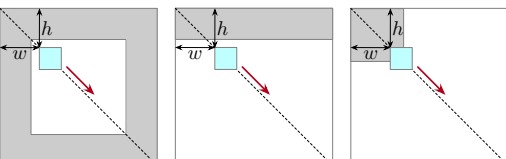

Figure 4: Illustration of trigger-prompt interactions by moving the trigger along the diagonal with position $(h, w)$ for visual prompts following three different templates: padding (left), stripe (middle), and patch (right).

Table 2: The impact of trigger-prompt interactions on the attack performance.

| Prompt | Position | Overlap | Distance | CA (%) | ASR (%) |
|---|---|---|---|---|---|
| Padding | (0, 0) | 784 | 0 | 54.01 | 100.00 |
| | (28, 28) | 108 | 0 | 54.24 | 99.95 |
| | (56, 56) | 0 | 26 | 52.59 | 84.08 |
| | (84, 84) | 0 | 54 | 53.77 | 17.76 |
| Stripe | (0,0) | 784 | 0 | 31.24 | 99.93 |
| | (49,49) | 0 | 19 | 31.28 | 68.07 |
| | (98,98) | 0 | 68 | 31.19 | 19.48 |
| | (147,147) | 0 | 117 | 30.43 | 20.85 |
| Patch | (0,0) | 784 | 0 | 33.74 | 99.98 |
| | (49,49) | 784 | 0 | 34.44 | 99.98 |
| | (98,98) | 0 | 18 | 32.78 | 21.64 |
| | (147,147) | 0 | 67 | 32.41 | 20.92 |

### 3.4 Improving Distant Triggers

The above analyses suggest that a successful attack requires placing the trigger at specific positions. This increases the possibility of detecting the trigger and further mitigating the attack (Wang et al., 2019; Cho et al., 2020). Therefore, here we explore simple solutions to improve the attack for distant trigger positions. We focus on the padding prompt with the trigger placed at the image center and evaluate RN50 on CIFAR10.

**Larger Trigger Size/Poisoning Ratio or Coefficient $\lambda$.** Based on the results in Section 3.2, a larger trigger size or poisoning ratio generally leads to better attack performance. However, we find that the ASR results are just around $33\%$ even when the trigger size is increased to $8 \times 8$ or the poisoning ratio is increased to $15\%$. According to the attack objective in Equa-

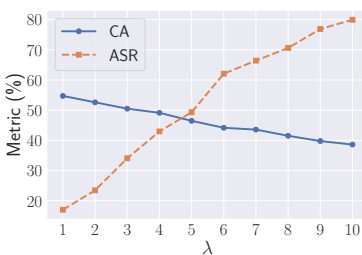

Figure 5: Improving distant triggers by increasing $\lambda$ in Equation 2.

tion 2, a straightforward way to improve the attack is to increase the coefficient $\lambda$. However, as can be seen from Figure 5, although a larger $\lambda$ leads to higher attack success, the model utility substantially decreases.

**Trigger Pattern Optimization.** Instead of simply fixing the trigger pattern as above, here we treat the trigger as a learnable variable. We follow the bi-level optimization (Huang et al., 2020; Geiping et al., 2021) to alternatively update the prompt and trigger using the same loss function in Equation 2. The detailed optimization procedure with hyperparameter selection is described in Appendix D. As can be seen from Table 3, the optimized triggers yield consistently high ASRs with little impact on CAs. For example, a small trigger size of $4 \times 4$ is sufficient to achieve a high attack performance above $85\%$ on average. Note that trigger optimization inevitably requires additional computations, so it makes little sense to apply it to our main experiments, where a fixed trigger already works perfectly.

Table 3: Improving distant triggers by trigger pattern optimization.

| Model | Prompt | Metric | Trigger Size | | Poisoning Ratio (%) | |
|---|---|---|---|---|---|---|
| | | | 2 | 4 | 10 | 15 |
| RN50 | Clean | CA (%) | 54.99 | 54.99 | 54.99 | 54.99 |
| | | ASR (%) | 15.15 | 26.60 | 15.14 | 15.16 |
| | Backdoor | CA (%) | 55.35 | 54.76 | 54.69 | 53.15 |
| | | ASR (%) | 28.33 | 61.56 | 46.92 | 62.26 |
| BiT-M | Clean | CA (%) | 61.91 | 61.91 | 61.91 | 61.91 |
| | | ASR (%) | 14.25 | 27.35 | 13.56 | 13.78 |
| | Backdoor | CA (%) | 62.29 | 62.22 | 63.08 | 61.92 |
| | | ASR (%) | 87.71 | 96.40 | 98.24 | 99.37 |
| Ins. | Clean | CA (%) | 64.22 | 64.22 | 64.22 | 64.22 |
| | | ASR (%) | 12.18 | 17.23 | 12.17 | 12.53 |
| | Backdoor | CA (%) | 63.20 | 66.15 | 64.55 | 62.82 |
| | | ASR (%) | 61.67 | 92.96 | 89.15 | 94.46 |
| CLIP | Clean | CA (%) | 92.94 | 92.94 | 92.94 | 92.94 |
| | | ASR (%) | 10.20 | 9.97 | 10.01 | 10.12 |
| | Backdoor | CA (%) | 93.13 | 92.86 | 93.06 | 93.26 |
| | | ASR (%) | 99.82 | 99.94 | 99.99 | 99.94 |

## 4 EXPERIMENTS OF DEFENSES

We evaluate six well-known model- and input-level backdoor defenses and also introduce a new, prompt-level detection approach that solely relies on the prompt features. Model-level defenses are applied to the prompted model, i.e., the combination of the frozen (clean) pre-trained model $M$ and the backdoored visual prompt $\mathbf{w}_b$. Since the pre-trained model is clean, we can induce that the visual prompt is backdoored if abnormal behaviors are shown. Note that dataset-level defenses (Chen et al., 2018; Tang et al., 2021) are not applicable to our scenario because the attacker (i.e., VPPTaaS) does not send any backdoored dataset but the backdoored visual prompt to the victim (i.e., downstream user). We focus this section on fixed triggers and leave similar experiments on optimized triggers to Appendix I. Note that their conclusions are very similar.

### 4.1 MODEL-LEVEL BACKDOOR DETECTION

**Trigger-Reconstruction-Based Detection.** Neural Cleanse (Wang et al., 2019) is a backdoor defense based on trigger reconstruction. The main idea is the minimum perturbation required to reconstruct the trigger for the backdoor target label should be substantially smaller than that for other labels. Given the reconstructed triggers for all labels, an anomaly index is calculated on the statistical distribution of the norm values of these triggers. When the anomaly index is larger than a threshold $T$, the model (the prompted model in our case) is treated as backdoored. To evaluate the effectiveness of Neural Cleanse, we generate 5 clean and 5 backdoored visual prompts on CIFAR10. We adopt the recommended threshold $T = 2$ and also other default parameter settings from the original work (Wang et al., 2019). We show the ROC curves together with AUC scores in Figure 6. The recommended threshold $T = 2$ leads to either low TPR (RN50, BiT-M, and Instagram) or high FPR (CLIP). We thus conclude that Neural Cleanse is not effective against our backdoor attacks.

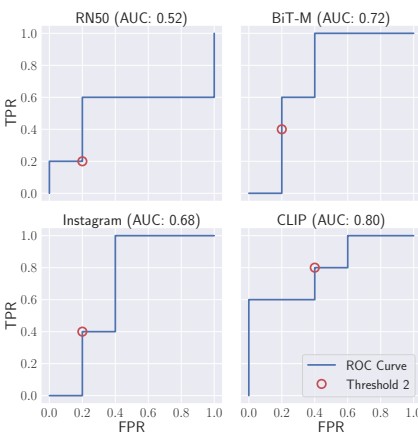

Figure 6: Backdoor detection by Neural Cleanse (Wang et al., 2019).

We further examine the trigger reconstruction results for both the failure and success cases for $T = 2$. Figure 7 visualize two such examples. For Figure 7a, the reconstruction is thought to fail since the anomaly index is 1.82 $< T$, but it indeed successfully locates the trigger at the bottom right corner. For Figure 7b, the reconstruction is

thought to be succeed since the anomaly index is $3.73 > T$, but it indeed fails to locate the trigger. We find that the above conflict happens half the time, suggesting that Neural Cleanse is not a reliable defense against our attack. Specifically, using only a scalar threshold may not sufficiently interpret the actual reconstruction results.

**Model-Diagnosis-Based detection.** MNTD (Xu et al., 2021) learns a meta binary detector to distinguish back-doored models from clean ones. It assumes a black-box access to the target model and as a result, the detector takes as input the output posteriors of a set of fine-tuned queries on the target model. We use the CIFAR10 dataset and CLIP model for evaluation purposes. For the backdoored visual prompts, we consider diverse poisoning ratios (i.e., $0.5\%$, $1\%$, $3\%$, $5\%$, and $10\%$), trigger sizes (i.e., from $1 \times 1$ to $5 \times 5$), and trigger locations (i.e., 9 in Figure 3a). In total, we obtain 340 backdoored visual prompts and the same number of clean visual prompts. Different random seeds are used to ensure that prompts generated in the same setting are different from each other.

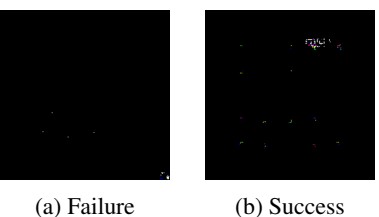

(a) Failure          (b) Success

Figure 7: Visualizations of reconstructed triggers for failure and success cases of Neural Cleanse.

For evaluation, we consider "known" and "unknown" scenarios. In the "known" scenario, we randomly sample $60\%$ of the above prompts for training and $40\%$ for testing. In the "unknown" scenario, we ensure the training and testing backdoored prompts are based on different parameters. Specifically, we select those generated with triggers located at the bottom right corner (180 in total) for training, and the rest 160 backdoored prompts for testing. The same number of clean visual prompts are used to ensure class balance. We find MNTD performs very well, with an area under the curve (AUC) score of 1.0 in the "known" scenario and 0.995 in the "unknown" scenario.

## 4.2 PROMPT-LEVEL BACKDOOR DETECTION

Since our backdoor is directly implanted into the visual prompt, it is worth exploring if it can be detected given only the prompt. To this end, we conduct similar experiments as in Section 4.1 but train a simple CNN detector containing 4 convolution layers instead of a meta detector in MNTD. This CNN detector takes as input the visual prompt combined with a pseudo-image full of zero pixel values. We find our prompt-level detection works perfectly, with the detection accuracy of $100\%$ in both the "known" and "unknown" scenarios.

We further examine the Grad-CAM saliency maps (Selvaraju et al., 2017) to interpret how our prompt-level detector works. As can be seen from Figure 8, besides being similarly effective, the two detectors for the two different scenarios also yield similar salient regions. Specifically, the salient regions spread the whole prompt for clean prompts but concentrate on local regions for backdoored prompts. This difference also confirms the perfect detection performance. Interestingly, for the backdoored visual

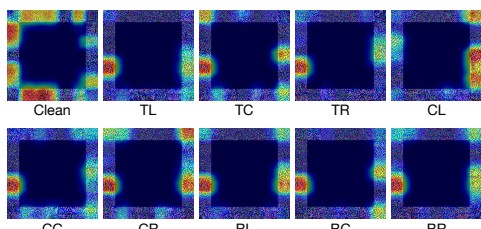

Figure 8: Grad-CAM visualizations of the clean vs. backdoored prompts (with triggers at 9 positions) for our prompt-level detector trained in the "known" scenario. Red regions correspond to high saliency scores. T, C, B, L, and R denotes Top, Central, Bottom, Left, and Right, respectively. Figure 13 in Appendix E further shows that the "unknown" scenario follows almost the same pattern.

prompts, the most salient regions do not overlap with the triggers, indicating that the backdoor information stored in the prompt is not around the trigger.

**MNTD vs. Our Prompt-Level Detector.** We further use the t-SNE (van der Maaten & Hinton, 2008) to help explain the good performance of MNTD and our prompt-level detector and compare their properties, as shown in Figure 9. We can observe that for both detectors, clean and backdoored prompts are clearly separable, confirming their good performance. A clear difference is that the MNTD samples are linearly separable but for our prompt-level detector, the clean prompts are densely

clustered and the backdoored ones surround this cluster. This may be explained by the fact that the dimension of pixel-space visual prompts is much higher than the output posteriors used in MNTD.

**Note on the Practicality.** Both MNTD and our prompt-level detector require a number of training prompts. In Figure 14 of Appendix F, we show that a stable and good detection performance requires around 60 training visual prompts. Similar to Carlini (2021), we argue that all these efforts, however, are infeasible for downstream users with limited resources. Otherwise, they do not need the VPPTaaS service in the first place. Therefore, detection-based defenses may not be practical in our scenario.

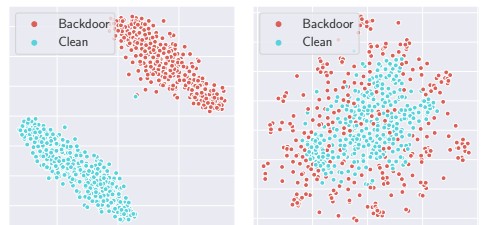

Figure 9: The t-SNE visualizations for MNTD trained in the "known" scenario (left) and our prompt-level detector (right).

### 4.3 INPUT-LEVEL BACKDOOR DETECTION

Backdoor detection is also commonly conducted at the input level, where clean inputs are accepted for further use but backdoored inputs are rejected. The detection performance is evaluated based on two metrics: False Rejection Rate (FRR) and False Acceptance Rate (FAR). FAR represents the percentage of backdoored inputs that are falsely detected as clean. FRR represents the percentage of clean inputs that are falsely detected as backdoored. A detection method is expected to achieve a low FAR for effectiveness and a low FRR for maintaining the model utility. We consider two detection methods, SentiNet (Chou et al., 2020) and STRIP (Gao et al., 2019). The intuition of SentiNet is that strong localized universal attacks usually cause the saliency of the pre-trained model to concentrate on the localized perturbations, e.g., the triggers in backdoor attacks. Model predictions on such strongly concentrated salient regions persist no matter how the rest image regions change. STRIP relies on a more general intuition that the model prediction on a backdoored input is more invariant to image corruptions than that on a clean input. See Appendix G for detailed descriptions.

For evaluating SentiNet, we first ensure that it is applicable in our case by showing that the saliency of the backdoored image indeed concentrates on the trigger in Appendix H. Then, we conduct quantitative experiments on 1,000 clean and 1,000 backdoored images. As can be seen from Table 4, SentiNet performs particularly badly for CLIP (i.e., FAR = 35.20%). We further check the salient regions generated for backdoored input images, and we find that in *all* false acceptance cases, Grad-CAM fails to locate the triggers accurately. On the other hand, SentiNet yields relatively high FRRs

Table 4: Backdoor detection results of SentiNet (Chou et al., 2020) and STRIP (Gao et al., 2019).

| Defense | Metric | Model | | | |
|---|---|---|---|---|---|
| | | RN50 | BiT-M | Ins. | CLIP |
| SentiNet | FAR (%) | 0.00 | 8.30 | 1.00 | 35.20 |
| | FRR (%) | 9.20 | 8.50 | 11.30 | 9.10 |
| STRIP | FAR (%) | 0.05 | 0.00 | 0.25 | 1.35 |
| | FRR (%) | 2.80 | 3.65 | 3.20 | 1.15 |

(around 10%), leading to a non-negligible drop in model utility. For evaluating STRIP, we use 2,000 clean inputs to determine the entropy threshold (by ensuring the FRR on these clean inputs is 1%). We then employ another 2,000 clean and 2,000 backdoored inputs for detection. A softmax function is used to process the output posteriors before the entropy calculation in our experiments. As can be seen from Table 4, STRIP is superior to SentiNet, especially for CLIP.

**Bypass Input-Level Detection.** Both SentiNet and STRIP require attacks to be strong so that the model prediction on a backdoored input is consistent over multiple overlaid images. Therefore, we further explore if the attacker can bypass SentiNet and STRIP by intentionally restricting their strength. Specifically, we adopt the $4 \times 4$ optimized trigger on the RN50 model in Section 3.4. We find this modification still yields an acceptable ASR (i.e., 61.56%) but drastically increases the FAR to 37.10% for SentiNet and 87.20% for STRIP. These results indicate that it is possible to largely compromise the performance of SentiNet and STRIP by adopting a moderate attack.

### 4.4 BACKDOOR MITIGATION

Although users can reject a backdoored model/prompt based on detection results, it may be impractical because finding another service provider requires additional resources and expertise (Wang et al.,

2019). In this case, backdoor mitigation is a promising alternative, which is normally achieved by eliminating backdoors from the model/prompt or trigger patterns from the input.

**Prompt-Level Backdoor Mitigation.** Fine-Pruning (Liu et al., 2018b) applies network pruning (Han et al., 2016; Molchanov et al., 2017) to mitigate model backdoors. Fine-Pruning iteratively prunes neurons with the lowest average activations on a validation dataset until a pre-defined pruning fraction $\eta$ is reached. In our case, we fix the model but prune the pixels with the lowest absolute values from the visual prompt. As can be seen from Figure 10, there is a clear trade-off between ASR and CA for RN50, BiT-M, and Instagram. For example, when ASR starts to substantially drop for $\eta > 70\%$, the CA also drops dramatically. In contrast, for CLIP, CA is consistently well maintained, even when $\eta = 100\%$, i.e., downstream users choose to completely drop the use of the visual prompt. This contrast might be explained by the strong "zero-shot" capability of CLIP, which makes CLIP easily transfer to other downstream tasks via simple natural language instructions without fine-tuning on downstream data (Radford et al., 2021).

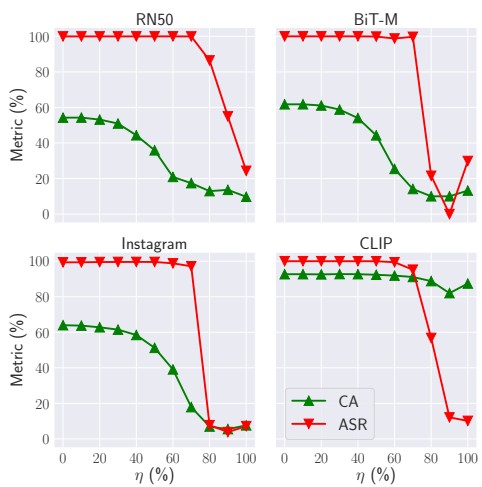

Figure 10: Backdoor mitigation by Fine-Pruning.

**Input-Level Backdoor Mitigation.** DAPAS (Cho et al., 2020) trains a Denoising AutoEncoder (DAE) (Vincent et al., 2010) on balanced clean and perturbed images and then uses this trained DAE to eliminate potential perturbations. We follow the settings of Cho et al. (2020) and report the results in Table 5. Here, "w/o" means no DAPAS is applied, while "Noise" means training DAPAS with Gaussian noise, and "Trigger" means an extreme scenario in which the defender constructs the perturbed training using the identical trigger to that of the attacker. We can observe that "Noise" is sufficient to

Table 5: Backdoor mitigation by DAPAS.

| DAPAS | Metric | Model | | | |
|---|---|---|---|---|---|
| | | RN50 | BiT-M | Ins. | CLIP |
| w/o | CA (%) | 54.24 | 61.75 | 63.94 | 92.66 |
| | ASR (%) | 99.96 | 100.00 | 99.36 | 99.99 |
| Noise | CA (%) | 16.57 | 17.20 | 17.64 | 40.65 |
| | ASR (%) | 0.00 | 0.10 | 0.21 | 4.84 |
| Trigger | CA (%) | 16.52 | 17.10 | 17.64 | 40.19 |
| | ASR (%) | 0.00 | 0.04 | 0.09 | 4.56 |

guarantee very low ASR results and "Trigger" further decreases them. However, DAPAS also trades off the CA by about $40\%$.

## 5 CONCLUSION AND OUTLOOK

We provide the first systematic study of security vulnerabilities of visual prompt learning (VPL) from the lens of backdoor attacks. From the attack perspective, we propose BadVisualPrompt, the first backdoor attack against VPL, and demonstrate its general effectiveness over different models, datasets, and VPL variants. We particularly analyze the impact of trigger-prompt interactions on the attack performance and show that the attack performance may be largely decreased when the trigger and prompt are distant. From the defense perspective, we demonstrate that representative detection- and mitigation-based methods are either ineffective or impractical against our BadVisualPrompt. We also provide new insights into their behaviors in both the success and failure cases. Although our new attack may be potentially misused by malicious actors, we firmly believe that our systematic analyses of the vulnerability of VPL can provide more to future studies for designing effective defenses.

Since visual prompt learning (VPL) just gets substantial attention very recently, it is understandable that VPL performs not well enough in certain settings. However, we can already see performance improvement in recent attempts (Chen et al., 2023b; Huang et al., 2023). Since we are the first to explore the vulnerabilities of VPL, we have focused our study solely on the well-defined, backdoor attacks. Moving forward, it would be necessary to explore other vulnerabilities. In addition, although we have tried our best to evaluate diverse defenses against our new attack, no effective defenses have been found so far. Future work should explore new defense strategies to defeat our attack.

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

# A  RELATED WORK

**Prompt Learning in NLP.** Prompt learning (Shin et al., 2020; Hambardzumyan et al., 2021; Lester et al., 2021; Li & Liang, 2021) is a new machine learning paradigm in natural language processing (NLP). The key goal is identifying prompts that allow a pre-trained language model to effectively perform downstream tasks. Current research in prompt learning can broadly be grouped into two categories. The first category is manual prompt engineering. Existing efforts in this category focus on manually creating prompts based on human intuition and domain knowledge of the downstream tasks, including template engineering (Petroni et al., 2019) and demonstration design (e.g., selection (Reynolds & McDonell, 2021) and ordering (Lu et al., 2022)). Similar to feature engineering in classical machine learning, identifying optimum manual prompts is non-trivial and the resulting prompts can be sensitive to small perturbations (Li & Liang, 2021). Recent efforts thus have focused on automated prompt engineering, which aims to automatically generate or optimize the prompts by learning from the input-output pairs. Representative methods include prompt mining (Jiang et al., 2020), prompt scoring (Davison et al., 2019), prefix turning (prompt tuning) (Li & Liang, 2021), hybrid tuning (Han et al., 2021), etc. Owing to their practicability and proven efficacy, methods in both categories have been successfully applied in many areas in NLP, including language models (Puri & Catanzaro, 2019; Reynolds & McDonell, 2021), machine translation models (Brown et al., 2020; Reynolds & McDonell, 2021), dialogue systems (Liu et al., 2018a; Lai et al., 2020), text classification (Puri & Catanzaro, 2019; Schick et al., 2020), and knowledge probing (Petroni et al., 2019; Brown et al., 2020).

**Visual Prompt Learning.** Inspired by prompt learning in NLP, visual prompt learning (Bahng et al., 2022; Gan et al., 2023; Wu et al., 2022; Chen et al., 2023a) also starts emerging in the computer vision domain to adapt the large-scale upstream pre-trained visual models to various downstream tasks, such as image classification (Bahng et al., 2022), vision-language modelling (Zhou et al., 2022), etc. The majority of efforts in visual prompt learning aim to learn a pixel perturbation that can be combined with the input image and allow the pre-trained model to complete the downstream tasks without model fine-tuning (Howard & Ruder, 2018). There also exists a different research direction, namely visual prompt tuning (Jia et al., 2022b), where the visual prompt is in the form of additional model parameters instead of pixel perturbations at the input space. Our study does not consider this front as the victim audits the pre-trained model for compliance purposes.

**Backdoor Attacks.** A backdoor attack is a training-time attack where the attacker alters the training data of a machine learning system to implant a backdoor by label manipulation (Gu et al., 2017; Yao et al., 2019; Du et al., 2022) and data manipulation (Saha et al., 2020; Zhao et al., 2020; Souri et al., 2022). The hidden behavior is triggered when a specific pattern is present in future inputs at the test time. Many backdoor attacks are proposed considering different attack scenarios (e.g., adopting third-party datasets (Liu et al., 2023), platforms (Wang et al., 2020), and models (Jia et al., 2022a; Shen et al., 2021)). Pending on the attack goals, the backdoor attacks can be grouped into targeted attacks (Jia et al., 2022a; Souri et al., 2022) and untargeted attacks (Luo et al., 2022; Kiourti et al., 2020). These attacks have been successfully launched against different machine learning paradigms, including supervised learning, self-supervised learning (Saha et al., 2022), and federated machine learning (Wang et al., 2020). Cai et al. (2022) have explored backdoor attacks against prompt learning but in the NLP domain. However, text prompts are usually discrete and so cannot be directly applied to the vision domain. Our work exemplifies that substantial efforts are required to understand/address the implications/challenges of various attack settings, especially the new challenge related to the interaction between the backdoor trigger and visual prompt, which does not exist in NLP.

**Backdoor Defenses.** To mitigate various backdoor attacks, several detection studies have been proposed to detect backdoors and can be roughly categorized into three levels, i.e., input-level (Gao et al., 2019; Ma et al., 2019), model-level (Wang et al., 2019; Xu et al., 2021) and dataset-level (Chen et al., 2018) defenses. Input-level detection methods (Gao et al., 2019; Ma et al., 2019) identify if the input data will trigger abnormal behaviors. Model-level detection methods (Wang et al., 2019; Xu et al., 2021) aim to infer whether the target model has been backdoored. Dataset-level detection methods (Chen et al., 2018; Tang et al., 2021) check whether the training dataset has been poisoned for backdoor attacks. Moreover, there are some other defense studies (e.g., Fine-Pruning Liu et al. (2018b)) proposed to mitigate or remove backdoor attacks when we already know that the target model is backdoored. However, these methods might also negatively affect the model utility of the

target model. In this paper, we explore both detection and mitigation methods at both the model and input levels, and we make further prompt-level defense attempts for our new scenario.

## B  DETAILED EXPERIMENTAL SETUPS

**Datasets.** We use three benchmark image datasets in our experiments, including CIFAR10 (CIF), SVHN (Netzer et al., 2011), and EuroSAT (Helber et al., 2018). The CIFAR10 dataset is a dataset consisting of 60,000 $32 \times 32$ images from 10 different classes of objects. The SVHN dataset is a dataset consisting of 99,289 $32 \times 32$ images from 10 different classes of street house numbers. The EuroSAT dataset is a dataset consisting of 27,000 $64 \times 64$ Sentinel-2 satellite images from 10 different classes of land use. We use the official training and testing data splits for the CIFAR10 and SVHN datasets. For the EuroSAT dataset, we randomly sample $80\%$ images per class for training and the rest $20\%$ for testing.

**Pre-trained Models.** We use four different pre-trained models for our experiments, including ResNet trained on ImageNet-1K (RN50) (He et al., 2016; Russakovsky et al., 2015), Big Transfer (BiT-M) (Kolesnikov et al., 2020), ResNeXt trained on 3.5B Instagram images (Instagram) (Mahajan et al., 2018), and CLIP (Radford et al., 2021). RN50, BiT-M, and Instagram are vision models. They accept an image as the input and output a prediction over a set of pre-defined classes. Differently, CLIP is a vision-language model. It takes an image and the names of all the classes (usually each class is in the form of a text prompt such as a sentence like "a photo of a [LABEL]") to form the set of potential text pairings and predict the most probable (image, text) pair. Table 6 provides an overview of these pre-trained models.

**Prompt Learning Settings.** We use the official PyTorch implementation of Bahng et al. (2022) to construct visual prompts.[2] We optimize each visual prompt on the downstream task for 100 epochs with a batch size of 128. We use stochastic gradient descent (SGD) as the optimizer and cross-entropy as the loss function. Specifically, we use SGD with a cosine scheduler. The initial learning rate is set to 40 with a momentum of 0.9. All images are resized to the desired input size of the pre-trained models ($224 \times 224$ in our study) before being combined with the visual prompt. Following the common practice (Bahng et al., 2022), we set the visual prompt as four-edge padding with a width of 30 pixels on each edge and map each pre-trained class index $i$ to downstream class index $i$ for the vision models while constructing a semantically similar text prompt for each downstream class for the CLIP model.

**Attack Settings.** Unless otherwise mentioned, we set the poisoning ratio to $5\%$ and place the trigger at the bottom right corner of the image. Since different image datasets may have different image sizes, we set the default trigger size as 1/16 of the original image size to compare the effectiveness of backdoor attacks on them fairly. As such, we set the default trigger size as $2 \times 2$ for CIFAR10 and SVHN, while $4 \times 4$ for EuroSAT. For the target class, we choose "automobile" for CIFAR10, "1" for SVHN, and "forest" for EuroSAT, all mapping to class index 1. We set $\lambda = 1.0$ in Equation 2 for our backdoor attack.

**Evaluation Metrics.** In this work, the pre-trained model is always used together with the visual prompt to give predictions on the downstream task. We use Clean Accuracy (CA) and Attack Success Rate (ASR) to measure the performance of our backdoor attack. Here CA represents the percentage of clean test images whose predicted labels are the same as the ground-truth labels, while ASR represents the percentage of backdoored test images whose predicted labels are the same as the target labels. Concretely, CA measures the model utility goal while ASR measures the attack effectiveness goal of our attack. In general, higher CA and ASR indicate a more effective backdoor attack.

**Computation Resources.** We run the experiments on an internal cluster with $2\times$ AMD Rome 7742 @ 2.25 GHz CPUs, 1 TB RAM, and $8\times$ NVIDIA DGX A100 GPUs. It takes around 7, 11, and 3 hours to fine-tune a visual prompt on the CIFAR10, SVHN, and EuroSAT datasets respectively. The computation time for generating an optimized trigger in Algorithm 1 mainly depends on $K$ and is about $K$ times the aforementioned fine-tuning time.

---

[2]https://github.com/hjbahng/visual_prompting.

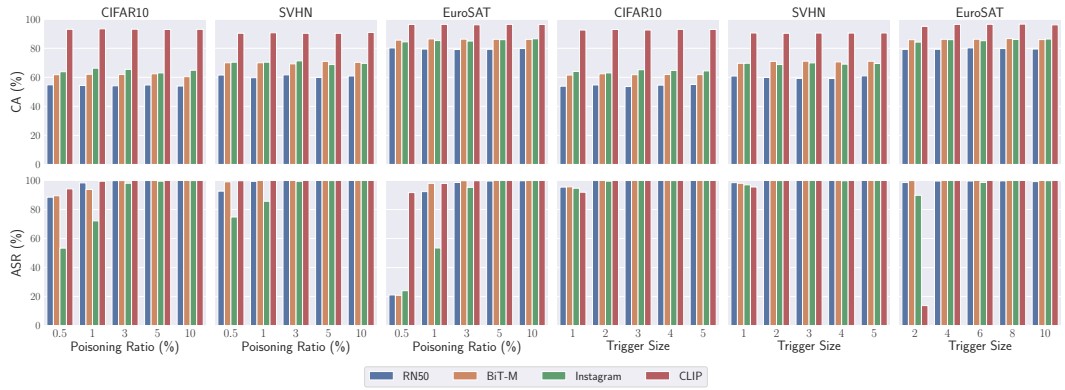

Figure 11: Impact of the poisoning ratio and trigger size on the attack performance.

Table 6: An overview of the pre-trained models.

Table 7: Impact of $K$ on attack performance of Algorithm 1.

| Model | Architecture | Modality | Pre-trained Dataset |
|---|---|---|---|
| BiT-M (Kolesnikov et al., 2020) | ResNet50 | Vision | 14M ImageNet-21K |
| RN50 (He et al., 2016) | ResNet50 | Vision | 1.2M ImageNet-1K |
| Instagram (Mahajan et al., 2018) | ResNext101 | Vision | 3.5B Instagram photos |
| CLIP (Radford et al., 2021) | ViT-B/32 | Vision-Language | 400M image-text pairs |

| Metric | $K$ | | | | |
|---|---|---|---|---|---|
| | 1 | 2 | 3 | 4 | 5 |
| CA(%) | 54.44 | 54.76 | 56.50 | 56.39 | 55.97 |
| ASR(%) | 58.09 | 61.56 | 58.98 | 64.86 | 63.07 |

## C  DETAILED ATTACK RESULTS WITH VARIED POISONING RATIOS AND TRIGGER SIZES

Here we provide detailed explanations for the difference in attack performance between CLIP and the vision models shown in Figure 11. CLIP is known to have higher capabilities than vision models in terms of transferability (Radford et al., 2021) and robustness (Fang et al., 2022). Such capabilities mainly rely on the quality rather than the quantity of the training data (Nguyen et al., 2022). When the trigger size is small (e.g., $2 \times 2$ for EuroSAT), the backdoored image is in "poor" backdoor quality. Therefore, CLIP has a low capability of learning the backdoor information, leading to a low ASR. However, when the trigger becomes large enough (e.g., the default $4 \times 4$ for EuroSAT), CLIP has a high capability, leading to a very high ASR even when the poisoning ratio is small.

The above phenomenon is especially obvious on EuroSAT, as shown in Figure 11. We conjecture this is related to the trigger complexity and data size. Specifically, to ensure the same ratio of the trigger size to the original image for different datasets, the smallest trigger size for EuroSAT is set to $2 \times 2$, while that for other datasets is $1 \times 1$. This causes the trigger pattern for EuroSAT (i.e., black and white) to be more complex than that for other datasets (only white). This consequently leads to even lower backdoor quality, as introduced above. We verify this conjecture by showing that attacking with a $2 \times 2$ trigger with only white pixels yields a high ASR, $98.65\%$, in contrast to the low ASR, $13.80\%$, of its black and white counterpart. On the other hand, since the EuroSAT dataset is relatively small, using a low poisoning ratio is not enough for vision models to learn the backdoor information.

## D  ADDITIONAL DETAILS OF OPTIMIZED TRIGGERS

The algorithm for our backdoor attack with trigger optimization is illustrated in Algorithm 1. We use an SGD optimizer with an initial learning rate of 1.0 and a cosine scheduler to update the trigger and set the outer updating steps $R = 100$. In particular, at each time of trigger update, the visual prompt is updated $K$ times. We first test the impact of $K$ on the attack performance with the pre-trained RN50 model and the $4 \times 4$ trigger size. The results are shown in Table 7. We can observe that the attack performance is not sensitive to the setting of $K$. In our work, we use $K = 2$ for a good trade-off between efficiency and attack effectiveness. We visualize the optimized triggers with different trigger sizes in Figure 12. Table 7 shows that the attack performance is not sensitive to the setting of the inner optimization step, $K$ in Algorithm 1.

---

**Algorithm 1:** Attack with Trigger Optimization

---

**Input:** Visual prompt $\mathbf{w}$, trigger $\Delta$, clean data $(\mathbf{x}, y) \in \mathcal{D}_{\text{clean}}$, target label $t$, poisoning ratio $p$,
        outer/inner step $R/K$

**Output:** Optimized trigger $\Delta$, backdoored visual prompt $\mathbf{w}$

1   Initialize $\Delta$ and $\mathbf{w}$;

2   Randomly sample $p$ proportion of $\mathcal{D}_{\text{clean}}$ as $\mathcal{D}_{\text{poison}}^*$;

3   **for** $i \leftarrow 1$ **to** $R$ **do**
         // Get the poisoned dataset with the current trigger $\Delta$

4       $\mathcal{D}_{\text{poison}} \leftarrow \left\{ (\mathbf{x}_{\text{poison}}, t) : \mathbf{x}_{\text{poison}} \leftarrow \mathcal{P}(\mathbf{x}, \Delta, t), \forall (\mathbf{x}, y) \in \mathcal{D}_{\text{poison}}^* \right\}$;
         // Optimize the visual prompt $\mathbf{w}$

5       **for** $j \leftarrow 1$ **to** $K$ **do**

6          |   Update $\mathbf{w}$ with Equation 2;
         // Optimize the trigger $\Delta$

7       Update $\Delta$ with the second term of Equation 2;

8   **return** $\Delta$ and $\mathbf{w}$

---

## E   GRAD-CAM VISUALIZATIONS FOR OUR PROMPT-LEVEL DETECTOR

Figure 13 visualizes the results for our prompt-level detector trained in the "unknown" scenario, in which the backdoor triggers in the inference phase have never been seen in the training phase, as proposed in Section 4.2. We can draw the same conclusion as for Figure 8 that the salient regions for the clean vs. backdoored prompts are different, and for the backdoored visual prompts, the most salient regions do not overlap with the trigger location.

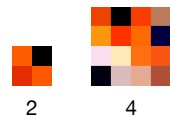

2         4

Figure 12: Optimized triggers with various trigger sizes.

## F   DETECTION WITH VARIED TRAINING DATA SIZES

Here we explore the impact of the training data size $N_{\text{train}}$ on the detection performance of the model-level detector, MNTD, and our prompt-level detector. The evaluation results under the "known" scenario are shown in Figure 14. We could observe that a larger training data size leads to better detection performance. Specifically, when the size is relatively small, both two detectors show large standard deviations, indicating that these two detectors are not reliable solutions to defend against our new attack.

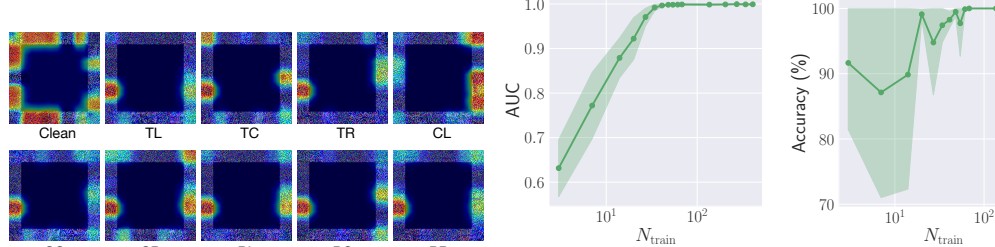

Figure 13: Grad-CAMs for our prompt-level detector in the "unknown" scenario. Red regions correspond to high saliency scores.

Figure 14: The impact of the training data size on MNTD (left) and our prompt-level detector (right). We repeat each experiment 10 times and visualize the standard deviations with the error bands around the curves.

## G  DETAILED DESCRIPTIONS OF SENTINET AND STRIP

SentiNet (Chou et al., 2020) is one typical input-level detection method that is based on saliency maps. The intuition of SentiNet is that strong localized universal attacks usually cause the saliency of the pre-trained model to concentrate on the localized perturbations, e.g., the adversarial patch (Brown et al., 2017) in adversarial attacks or the trigger pattern in our studied backdoor attacks. Model predictions on such strongly concentrated salient regions persist no matter how the rest image regions change.

Based on this intuition, for each input, SentiNet first determines the salient regions with saliency scores higher than the 15% of the maximum value on the Grad-CAM saliency map. The content of these salient regions on the input image is regarded as the extracted pattern. It then overlays the extracted pattern on multiple (100 by default) clean images to see whether the model predictions remain the same. SentiNet also considers the non-malicious overlay scenario in which the benign patterns usually cause few misclassifications with high prediction confidence while the inert patterns with low saliency (e.g., Gaussian noise) often occlude objects and thus disturb the model, resulting in low prediction confidence. Therefore, SentiNet utilizes two features to distinguish malicious patterns from benign ones, one is the number of images overlaid with the extracted pattern that successfully fool the target model, and the other is the average prediction confidence of the target model when we replace the extracted pattern content with random noise. Specifically, SentiNet calculates the above features for 400 clean data points to fit a 2-dimensional boundary curve and estimates a threshold by the distances of clean images lying outside the boundary. In the inference time, SentiNet calculates the above features for any testing image to get the corresponding distance to the boundary. If the resulting distance is above the pre-calculated threshold, the input image is identified as backdoored.

STRIP (Gao et al., 2019) follows a similar idea to SentiNet but eliminates the need for generating saliency maps. Instead, it relies on a more general intuition that the model prediction on a backdoored input is more invariant to image corruptions than that on a clean input. Specifically, STRIP randomly samples $N$ (100 by default) clean images and overlays each of them to the input to form a corrupted image set for this specific input. Then, STRIP computes the average entropy on the output posteriors of all overlaid images in the corrupted image set as the feature for the input image. STRIP collects the above entropy features for a set of

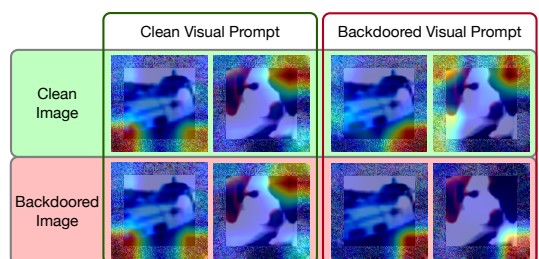

Figure 15: Grad-CAMs in SentiNet for clean/backdoored visual prompts and images.

seen clean images to determine a statistical threshold. For detection, an input is recognized as backdoored when its entropy is larger than the pre-computed threshold and otherwise as clean.

## H  GRAD-CAM VISUALIZATIONS FOR SENTINET

Before conducting quantitative experiments on evaluating SentiNet, we first verify if SentiNet is intuitively applicable in our case. To this end, we check if the saliency of the backdoored image indeed concentrates on the trigger with the pre-trained RN50 model. We visualize the Grad-CAM saliency maps for a "ship" image and a "dog" image in Figure 15. The backdoored image contains a $2 \times 2$ trigger at the bottom right corner, and the target class is set as "automobile". We make sure the pre-trained model misclassifies the images into "automobile" only when both the image and the prompt are backdoored. As can be seen, when the backdoored visual prompt is applied, the saliency of the backdoored images concentrates on the trigger location. We also notice that when the clean prompt is added, the saliency maps for the clean and backdoored images are very similar.

## I  DEFENSES AGAINST OPTIMIZED TRIGGERS

In addition to the defense experiments against fixed triggers conducted in Section 4, here we evaluate backdoor defenses against optimized triggers. In all cases, we set the trigger size as $4 \times 4$ and the

poisoning ratio as 5%. The trigger is placed at the center of the original image. If not mentioned specifically, all experiments follow the same settings to those in Section 4. In general, we find that all conclusions are consistent with those in Section 4.

**Model-Level Backdoor Detection.** Different from the main experiments on fixed triggers, here we only consider 9 different trigger positions but with the trigger size and the poisoning ratio fixed as default since optimizing triggers requires larger computations. In this case, we generate a total of 180 backdoored visual prompts and also 180 clean visual prompts. As can be seen from Figure 16, Neural Cleanse is still not effective in detecting the backdoored visual prompts. For MNTD detection, since generating a number of visual prompts for optimized triggers is more expensive, different from Section 4, here we fix the trigger size and poisoning ratio during generation. Accordingly, for the "unknown" scenario, we use backdoored visual prompts at six locations (i.e., BL, BC, BR, CL, CC, and CR) for training because training data at only one location are not enough. The AUC scores for the MNTD under both "known" and "unknown" scenarios are 1.0, indicating its effectiveness.

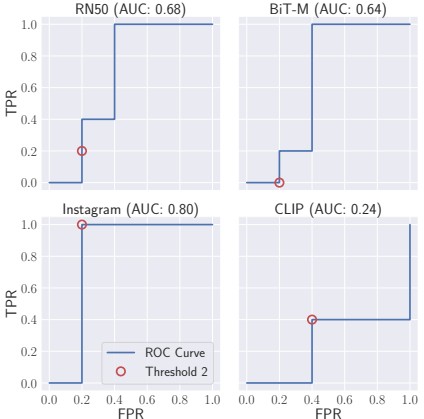

Figure 16: Backdoor detection by Neural Cleanse (Wang et al., 2019) for optimized triggers.

Figure 17: Backdoor mitigation by Fine-Pruning (Liu et al., 2018b) for optimized triggers.

**Prompt-Level Backdoor Detection.** Our prompt-level detector, which is introduced in Section 4.2 is still very effective, with an accuracy of 1.0 for both the "known" and "unknown" scenarios. However, as discussed in Section 4.2, both MNTD and our prompt-level detectors require a number of visual prompts, which is not realistic for the downstream user with limited resources.

**Input-Level Backdoor Detection.** We then conduct detection with SentiNet and STRIP) experiments following Section 4.3. The detection results are shown in Table 8. As can be seen from Table 8, SentiNet and STRIP become much worse compared to their performance on fixed triggers shown in Table 4. Specifically, SentiNet is consistently ineffective since it cannot accurately locate the trigger with Grad-CAM. STRIP is not effective when the attack is relatively weak (i.e., on RN50).

Table 8: Backdoor detection by SentiNet (Chou et al., 2020) and STRIP (Gao et al., 2019) for optimized triggers.

| Defense | Metric | Model | | | |
|---|---|---|---|---|---|
| | | RN50 | BiT-M | Instagram | CLIP |
| SentiNet | FAR (%) | 37.10 | 63.80 | 74.40 | 99.70 |
| | FRR (%) | 12.20 | 8.10 | 8.70 | 6.50 |
| STRIP | FAR (%) | 87.20 | 5.00 | 7.75 | 3.30 |
| | FRR (%) | 2.65 | 2.90 | 3.25 | 1.15 |

Table 9: Backdoor mitigation by DAPAS (Cho et al., 2020) for optimized triggers.

| DAPAS | Metric | Model | | | |
|---|---|---|---|---|---|
| | | RN50 | BiT-M | Instagram | CLIP |
| w/o | CA (%) | 54.75 | 62.22 | 66.11 | 92.86 |
| | ASR (%) | 61.59 | 96.41 | 92.97 | 99.94 |
| Noise | CA (%) | 16.03 | 21.35 | 16.81 | 42.52 |
| | ASR (%) | 6.03 | 0.26 | 0.18 | 53.42 |
| Trigger | CA (%) | 15.82 | 21.51 | 16.67 | 40.09 |
| | ASR (%) | 5.87 | 0.25 | 0.34 | 24.38 |

**Backdoor Mitigation.** As can be seen from Figure 17, the prompt-level mitigation method, Fine-Pruning, is only effective for CLIP but not for vision models, consistent with our finding on fixed triggers (see Figure 10). As can be seen from Table 9, the input-level mitigation method, DAPAS, is also not effective since it substantially sacrifices the model utility, consistent with our finding on fixed triggers (see Table 5).

