# OpenReview forum: "Prompt Backdoors in Visual Prompt Learning"
_ICLR.cc/2024/Conference — ICLR 2024 Conference Withdrawn Submission_

### Official Review · Reviewer_v9j8 · 2023-10-31

**Soundness:** 2 fair
**Presentation:** 3 good
**Contribution:** 1 poor
**Rating:** 3
**Confidence:** 4

**Summary:**

The paper proposes BadVisualPrompt backdoor attack method in Visual Prompt Learning task. The way to attack is construct poisoned data (attaching trigger $\Delta$ to random selected images) and optimize a visual prompt during training. The paper claims that poisoning 5% training data leads to 99% ASR with limited clean accuracy drop. Analysis of different backdoor defenses against proposed BadVisualPrompt is present.

**Strengths:**

1. Applying backdoor attack into Visual prompt learning is very interesting.

2. The paper is well-written and easy to understand.

3. The paper conduct both attack and defense practices, which is appreciable.

4. some analysis helps understand the effect of backdoor attack in visual prompt learning task.

**Weaknesses:**

1. In abstract, the author claimed "we identify and then address a new technical challenge related to interactions between the backdoor trigger and visual prompt". It is a bit vague what is the challenge. Please address it.

2. For attacking, the proposed BadVisualPrompt **a.** attaches the trigger ($\Delta$, with small patch with iterative white and black colors placed at the corner) to poisoned images, and **b.** optimizes backdoored Visual Prompt. I have two comments here:
+ 1) The b.optimized pattern is a traditional way in backdoor attack (e.g., UAP). The idea is not surprising.
+ 2) How important is the operation b? We know that simply attaching the trigger $\Delta$ is already sufficient in traditional backdoor attack secario (i.e., image classification in BadNet). Whether solo Operation *a* is already sufficient for a backdoor attack? In another word, the ablation study is needed.

3. In Section3.3, Figure3 and Table2, the author investigates the impact of trigger position. It seems as long as the trigger is inside/overlap the visual prompt, the ASR is high. Can author provide some insights why this happen? My understanding is related to Weaknesses 2: the trigger $\Delta$ is already sufficient enough for attack. The reason that there is low ASR when the trigger is not overlap with Visual prompt, may because involving the visual prompt during training in VPL task to make sure pre-trained label maps to downstream label.

4. There are a lot of backdoor attack baselines in general visual task (i.e., image classification). Is it possible to apply those baselines to the image? You can simply apply attack on images, without touching the visual prompt. Would the baseline attack be already sufficient?

5. The paper investigates both attack and defense, as the paper title "PROMPT BACKDOORS IN VISUAL PROMPT LEARNING". However, it looks like neither attack or defense is strong/solid enough.

**Questions:**

1. In Section 3.4, regarding larger trigger size, the author claims that "when the trigger size is increased to 8 x 8 or the posioning ratio is increased to 15%", the ASR is only around 0.33. Why this happen? seems not natural in default backdoor attack setting. (Figure 5 makes sense, that increasing lamba leads to increase ASR and decrease CA.)

---

> ### Author Response · Authors · 2023-11-17
> **Rebuttal by Authors**
>
> **Q1: Challenge mentioned in Abstract**
>
> **A1:** We find that our original attack method tends to have poor performance if we place the patch trigger far away from the visual prompt. Therefore, we propose several strategies to cope with this challenge. We will make it clearer in our paper.
>
> **Q2: The trigger optimization strategy is not surprising**
>
> **A2:** Since this paper is the first work to investigate the backdoor threats of visual prompts, we believe it is appropriate and necessary to apply the most established method proposed in the literature. Our major goal is not to propose new attack techniques, instead, we aim to rigorously quantify the risks of visual prompt learning with regard to backdoor attacks.
>
> **Q3: Whether merely using the trigger is enough for the backdoor attack**
>
> **A3:** This question seems to be based on a misunderstanding of backdoor attacks. The backdoor attack will only be activated when the input image with the trigger is combined with an optimized **backdoored** base, i.e., a model in conventional work or a visual prompt in our work. We can observe from Table 1 that the ASRs for clean visual prompts (with the backdoored input images) are much lower than those for backdoored ones, indicating the necessity of both a and b operations for a successful attack.
>
> **Q4: Explanation for the impact of trigger positions on attack performance**
>
> **A4:** This phenomenon may be because the model pays less attention to more distant locations from the visual prompt. This is evident by the visualizations in Figure 15 of Appendix H showing that the salient areas of (both clean and backdoored) prompted images are located around the visual prompt. We will add the explanations to our paper.
>
> **Q5: Applying previous methods without touching the visual prompt as baselines**
>
> **A5:** Obviously this is also based on a misunderstanding of backdoor attacks. The only component that memorizes the backdoor information in the attack process is the visual prompt, which will activate the backdoor behavior in the inference phase only when the input image contains the backdoor trigger. Therefore, the baseline methods must optimize the visual prompt in the attacking process. It seems that the reviewer wants to explore the adversarial patch against VPL. However, this is different from backdoor attacks and is beyond the research scope of this paper.
>
> **Q6: Neither attack nor defense is strong/solid enough**
>
> **A6:** We respectfully disagree. Our attack can achieve nearly a 100% ASR with negligible impact on model utility when inserting 5% poisoning samples. How could the attack performance be further promoted to be so-called “strong/solid”? Moreover, we have evaluated the performance of seven defense strategies that cover major representative defense categories against backdoor attacks. Some of them even show good performance (e.g., MNTD). So why does the reviewer deem the attack and defense not “strong/solid” enough?
>
> **Q7: Why increasing the trigger size or poisoning ratio does not work well in Section 3.4?**
>
> **A7:** You are right that the increase is limited, which is exactly why we further propose other strategies to address this problem.

---

### Official Review · Reviewer_ugXM · 2023-10-31

**Soundness:** 3 good
**Presentation:** 3 good
**Contribution:** 2 fair
**Rating:** 5
**Confidence:** 4

**Summary:**

The authors propose to implant a backdoor in visual prompt learning (VPL), which was suggested in some recent works as an alternative method to fine-tuning a large pre-trained computer vision model by adding learned perturbations to input data. Both attack and defense experiments were conducted to justify the effectiveness of the proposed backdoor approach.

**Strengths:**

S1. The experiments are comprehensive and mostly convincing. Specifically, the authors conducted experiments to explore both attack and defense strategies related to backdoors in visual prompts and analyzed the security risks associated with VPL.

S2. From an attack perspective, the authors demonstrated the effectiveness of the attack given a low poisoning ratio and a small trigger size.

S3. From a defense perspective, the authors highlighted the risk of the attack by showing that no effective defenses have been found so far.

**Weaknesses:**

W1. Weak motivation. The paper assumes that the Visual Prompt as a Service (VPPTaaS) provider is malicious. However, the claim that visual prompts are more resource-feasible can reduce the incentive for a user to rely on a VPPTaaS provider for fine-tuning a model.

W2. Lack of comparison with non-VPL methods, such as implanting the backdoor directly into the training images itself rather than the prompts.

**Questions:**

In addition to the weakness mentioned above, I wonder why a VPPTaaS provider would insist on injecting a backdoor into VPL instead of normal fine-tuning models, considering a more straightforward usage for the end-users in the latter case. Is there any industrial case to support your claim?

---

> ### Author Response · Authors · 2023-11-17
> **Rebuttal by Authors**
>
> **Q1: Weak motivation**
>
> **A1:** The VPPTaaS can help non-expert users adapt to the visual prompt learning paradigm. Compared to traditional third-party model finetuning services, VPPTaaS can adapt to various tasks with much less computation cost by running a single frozen backend model, thereby allowing selling its service at much lower prices. Some users with very limited resources cannot even run the large vision models locally, they can get the visual prompt from VPPTaaS and then query the service with the prompted image to get the final predictions. We will clarify our motivation more clearly in our paper.
>
> **Q2: Lack of comparison with non-VPL methods**
>
> **A2:** This seems to be a misunderstanding of backdoor attacks. The backdoor behavior will only be activated when both the backdoored visual prompt and the input image with the trigger appear together. The so-called “implanting the backdoor directly into the training images” is more like an adversarial attack, which is not relevant to our work. Furthermore, implementing other non-VPL methods in the VPL setting is meaningless since the model is fixed.
>
> **Q3: Why inject a backdoor into VPL instead of normal fine-tuning models**
>
> **A3:** We assume that the VPPTaaS service provider allows the victim to audit the entire backend pipeline (including the backend model) for security, privacy, and compliance purposes. And the users usually tend to choose trustworthy pre-trained models to serve as the backend models. In this case, modifying the model parameters of the pre-trained model will break the validated security (e.g., hash values) through the original auditing process. Additionally, backdooring the normal finetuning models has been well-studied in previous work. We will elaborate more on the motivation in our paper.

---

### Official Review · Reviewer_jfFZ · 2023-11-01

**Soundness:** 3 good
**Presentation:** 3 good
**Contribution:** 2 fair
**Rating:** 3
**Confidence:** 4

**Summary:**

This paper provides the first study of security pixel-space visual prompts learning (VPL) from the lens of backdoor attacks. The proposed backdoor attack method the authors proposed, BadVisualPrompt is the first backdoor attack against VPL. They conduct experiments to analyze the impact of trigger-prompt interactions on the attack performance. Besides the attack aspect, they also conduct experiments on existing backdoor detection and mitigation technologies on BadVisualPrompt and show they are either ineffective or impractical. They provide findings and analysis that might inspire further security research in VPL.

**Strengths:**

This paper is the first study to investigate the security of VPL against backdoor attacks.

The authors conduct extensive experiments to analyze the effect of hyperparameters and show that the distance between the trigger and prompt has a huge influence on the attack's effectiveness.

The paper both investigates backdoor attacks and existing defense.

**Weaknesses:**

Lack of novelty and challenge. While this paper is the first study of the backdoor attack on the visual prompt, the methodology largely mirrors the data-poisoning backdoor attack approach from BadNet[3]. This technique has already been extensively explored in the context of NLP prompts[5]. The readers may expect to see some challenges that arise with visual prompts learning, but the results show that directly using the conventional data-poisoning method can achieve a successful backdoor attack.

The presented threat model lacks widespread applicability. While platforms like PromptBase are popular for sharing text prompts in language tasks, it would be beneficial for the authors to highlight existing VPPTaaS platforms that share visual prompts.

Research is limited to pixel-space prompts. The study focuses solely on backdoor attacks targeting pixel-space visual prompts, neglecting the token-space visual prompts mentioned in [1,2]. Expanding the research to encompass attacks on token-space visual prompts would provide a more holistic view of the vulnerabilities.

[1] Menglin Jia, Luming Tang, Bor-Chun Chen, Claire Cardie, Serge Belongie, Bharath Hariharan, and Ser-Nam Lim. Visual Prompt Tuning. In European Conference on Computer Vision (ECCV). Springer, 2022b

[2] Gan, Y., Bai, Y., Lou, Y., Ma, X., Zhang, R., Shi, N., & Luo, L. (2023). Decorate the Newcomers: Visual Domain Prompt for Continual Test Time Adaptation. Proceedings of the AAAI Conference on Artificial Intelligence, 37(6), 7595-7603. https://doi.org/10.1609/aaai.v37i6.25922

[3] Gu, Tianyu, Brendan Dolan-Gavitt, and Siddharth Garg. "Badnets: Identifying vulnerabilities in the machine learning model supply chain." arXiv preprint arXiv:1708.06733 (2017).

[4] Chen, Xinyun, et al. "Targeted backdoor attacks on deep learning systems using data poisoning." arXiv preprint arXiv:1712.05526 (2017).

[5] Du, Wei, et al. "Ppt: Backdoor attacks on pre-trained models via poisoned prompt tuning." Proceedings of the Thirty-First International Joint Conference on Artificial Intelligence, IJCAI-22. 2022.

**Questions:**

Same as Weakness 2, are there any existing VPPTaas to sell and buy pixel-wise vision prompts?

The authors investigate the influence of the trigger location. The results in Figure 3 show that the trigger in the middle location has a suboptimal attack success rate. So what will happen if the attacker uses a pattern trigger[4] covered on the whole image rather than a patch trigger[3] in the specific location?

The reviewer would consider increasing the score after viewing the rebuttal responses.

**Details Of Ethics Concerns:**

-

---

> ### Author Response · Authors · 2023-11-17
> **Rebuttal by Authors**
>
> **Q1: Lack of novelty and challenge**
>
> **A1:** Since this is the first work to investigate this problem, we believe it is appropriate and necessary to apply the most established method proposed in the literature. Also, note that most of the previous works follow this method too. Our goal is not to propose new attack methods, instead, we aim to rigorously quantify the risks of visual prompt learning with regard to backdoor attacks. Moreover, our experiments show that our method already achieves a very strong performance, this further indicates the vulnerability of visual prompt learning. Furthermore, we identify and then address a new technical challenge related to interactions between the backdoor trigger and visual prompt, which does not exist in conventional, model-level backdoors.
>
> **Q2: The presented threat model lacks widespread applicability**
>
> **A2:** An optimal visual prompt is critical to the VPL performance and normally needs considerable efforts to optimize. Similar to applications in the NLP domain, VPPTaaS is promising to assist non-expert users in adapting to this new paradigm. Compared to traditional third-party model finetuning services, VPPTaaS can adapt to various tasks with much less computation cost by running a single frozen backend model, thereby allowing selling its service at much lower prices. Some service providers like Landing AI have already provided visual prompting services in various scenarios. We will clarify this more clearly and give some examples in our paper.
>
> **Q3: Research is limited to pixel-space prompts**
>
> **A3:** We assume that the pre-trained model is publicly available and has undergone rigorous security validation to audit their whole backend pipeline for security purposes. However, Jia et al. [1] form the prompt as additional model parameters instead of pixel perturbations at the input space. The parameter modifications introduced by this paradigm break the validated security through the original auditing process. In addition, backdooring such additional model parameters is conceptually similar to backdooring original model parameters as in conventional work. Therefore, this paradigm and its variants are out of our research scope.
>
> Gan et al. [2] utilize visual prompts for continual test time adaption, which is different from our attack scenario. They update the visual prompts for each query in the test time. However, in our work, the VPPTaaS optimizes the visual prompt only once and the visual prompt is fixed during the test time, so the user does not need to pay for the extra computational overhead caused by each query in the test time.
>
> **Q4: Existing VPPTaaS to sell and buy pixel-wise vision prompts**
>
> **A4:** Though currently there is no platform to trade pixel-wise visual prompts, some service providers like Landing AI have already provided the visual prompting service. It is very promising that such platforms will appear in the future when more powerful (and trustworthy) large vision models are released.
>
> **Q5: A pattern trigger covered on the whole image**
>
> **A5:** We use a cartoon image similar to Chen et al.’s work [4] to serve as the trigger pattern and conduct the backdoor attack with the blended injection strategy for the RN50 model on three datasets. The ASRs on all three datasets are higher than 92%, indicating the effectiveness of the backdoor attack. In this case, the trigger pattern always overlaps with the visual prompt and the minimum distance between the trigger and the visual prompt is zero.

---

### Official Review · Reviewer_jCHm · 2023-11-03

**Soundness:** 3 good
**Presentation:** 3 good
**Contribution:** 2 fair
**Rating:** 5
**Confidence:** 4

**Summary:**

The paper performs a systematic study of the security vulnerability of visual prompt learning. An attacker can optimize visual prompts such that when inputs have a trigger and the optimized prompt, this combination causes an intended misclassification. To achieve this attack objective, the paper proposes a data poisoning attack. The paper evaluates the effectiveness of the attack in different settings and shows the attack's success (when the trigger is not in the center of visual inputs). It is not also shown as easy to defeat the attack by existing defenses.

**Strengths:**

1. The paper presents a backdoor vulnerability of visual prompt learning.
2. The paper runs comprehensive evaluations in various settings.

**Weaknesses:**

1. It is not that surprising that the backdoor attack works on VPL.
2. It is also not technically surprising that one can compose such a crafting objective.

**Note:** I reviewed this paper from other conferences. I compared this paper from the previous submission and the current submission, and unfortunately, the major concern that quality reviewers raised has not been resolved in the paper. Thus, I am leaning toward borderline rejection.

> To me, given the vast literature on backdoor attacks, this paper seems to be a trivial extension of existing attacks to visual prompt learning. It means that even if the attack works, the fact that "the attack works" is not interesting. So do the results.


[Significance]

The attack is quite similar to existing backdooring. It is equivalent to optimizing a backdoor trigger under a constraint---a visual prompt. It can also be seen as constructing a universal adversarial patch with a backdoor trigger and a visual prompt.

I am also unclear about the practical importance of visual prompts. It's clearly motivated in natural language processing domains, but I am not sure about the goals of using this visual prompt in computer vision domains.


[Technical Novelty]

From another perspective, the paper can be novel when there are some new challenges in backdooring visual prompts. This can add knowledge transferable to attacking similar systems/models. But I couldn't find new challenges (and also the technical proposals to address them). This new backdoor attack just uses data poisoning attacks to achieve the goal.

**Questions:**

My questions are in the detailed comments in the weakness section.

**Details Of Ethics Concerns:**

No concern about the ethics

---

> ### Author Response · Authors · 2023-11-17
> **Rebuttal by Authors**
>
> **Q1: The result is not surprising**
>
> **A1:** We can only conclude that VPL is vulnerable to backdoor attacks after our comprehensive and systematic evaluations since we are the first to conduct this research. Technically, backdooring pixel-space visual prompt and backdooring models are different because the prompt has much fewer controllable parameters. Additionally, we explore a new angle that does not ally to conventional backdoors, i.e., trigger-prompt interactions, and we find and then address a specific challenge there.
>
> **Q2: Significance of this work and visual prompts**
>
> **A2:** The major contribution of this paper is not to propose new attack methods (since almost all backdoor attack methods are variants of the simple yet effective BadNet method), but the systematic and empirical evaluation of the security risks of visual prompt learning, a brand new learning paradigm for large vision models. This is a common type of research (representative papers are [a, b]). The backdoor attack is a fundamentally different attack from the adversarial attack, which can be based on adversarial patches. Visual prompt learning is important in the computer vision domain. Similar to the text prompt in the NLP domain, the visual prompt is an efficient approach to adapt the pre-trained large vision models to downstream tasks without finetuning the pre-trained models, thus significantly reducing the computation cost. Its significance has been well-recognized by the community. For instance, CVPR 2023 has accepted about 10 papers discussing various theoretical/practical aspects of visual prompt learning. Andrew Ng has also recently discussed the potential revolution in computer vision initiated by visual prompting. Some service providers like Landing AI have already provided access to visual prompting services under various scenarios. We will clearly clarify this in our paper.
>
> [a] Nicholas Carlini and Andreas Terzis. Poisoning and Backdooring Contrastive Learning. ICLR 2022.
>
> [b] Wei Du, Yichun Zhao, Boqun Li, Gongshen Liu, and Shilin Wang. PPT: Backdoor Attacks on Pre-trained Models via Poisoned Prompt Tuning. IJCAI 2022.
>
> **Q3: Technical novelty**
>
> **A3:** Since this is the first work to investigate this problem, we believe it is appropriate and necessary to apply the most established method proposed in the literature. Also, note that most of the previous work follows this method too. Our goal is not to propose new attack techniques, instead, we aim to rigorously quantify the risks of visual prompt learning with regard to backdoor attacks. Moreover, our experiments show that our method already achieves a very strong performance, this further shows the vulnerability of visual prompt learning.